# Intramedullary Spinal Cord Tumors: Whole-Genome Sequencing to Assist Management and Prognosis

**DOI:** 10.3390/cancers16020404

**Published:** 2024-01-18

**Authors:** Miguel Mayol del Valle, Bryan Morales, Brandon Philbrick, Segun Adeagbo, Subir Goyal, Sarah Newman, Natasha L. Frontera, Edjah Nduom, Jeffrey Olson, Stewart Neill, Kimberly Hoang

**Affiliations:** 1Department of Neurosurgery, Emory University Hospital, 1365 Clifton Road NE, Suite B6200, Atlanta, GA 30322, USA; sarah.newman2@emory.edu (S.N.); edjah.k.nduom@emory.edu (E.N.); jolson@emory.edu (J.O.); kimberly.bojanowski.hoang@emory.edu (K.H.); 2Department of Neuropathology, Emory University Hospital, 1364 Clifton Road, NE Room H-184, Atlanta, GA 30322, USA; bmoral3@emory.edu (B.M.); sgneill@emory.edu (S.N.); 3Department of Neurosurgery, Emory University School of Medicine, 100 Woodruff Circle, Atlanta, GA 30322, USAoadeagb@emory.edu (S.A.); 4Biostatistics Shared Resource Department, Winship Cancer Institute, Emory University, 1365-C Clifton Road, NE, Atlanta, GA 30322, USA; subir.goyal@emory.edu; 5School of Medicine, University of Puerto Rico Medical Sciences Campus, P.O. Box 365067, San Juan 00936-5067, Puerto Rico; natasha.frontera@upr.edu

**Keywords:** intramedullary spinal cord tumors, whole-genome sequencing, ATRX, p53, BRAFV600E, copy neutral loss of heterozygosity, glioblastoma

## Abstract

**Simple Summary:**

Intramedullary spinal cord tumors (IMSCTs), a rare category of neoplastic growth, comprise around two to five percent of tumors. Genetic analysis and sequencing to identify mutations can affect prognostication and management of these tumors. The aim of our retrospective analysis was to discern genetic alterations and describe the potential utility of genetic markers in the characterization of these tumors, thereby facilitating individualized therapy. In our cohort of eight patients undergoing whole-genomic sequencing, we suggest that loss of heterozygosity (LOH) is a genetic predictor of shorter progression-free survival in ependymomas and glioblastomas.

**Abstract:**

Intramedullary spinal cord tumors (IMSCTs) harbor unique genetic mutations which may play a role in prognostication and management. To this end, we present the largest cohort of IMSCTs with genetic characterization in the literature from our multi-site institutional registry. A total of 93 IMSCT patient records were reviewed from the years 1999 to 2020. Out of these, 61 complied with all inclusion criteria, 14 of these patients had undergone genetic studies with 8 undergoing whole-genomic sequencing. Univariate analyses were used to assess any factors associated with progression-free survival (PFS) using the Cox proportional hazards model. Firth’s penalized likelihood approach was used to account for the low event rates. Fisher’s exact test was performed to compare whole-genome analyses and specific gene mutations with progression. PFS (months) was given as a hazard ratio. Only the absence of copy neutral loss of heterozygosity (LOH) was shown to be significant (0.05, *p* = 0.008). Additionally, higher risk of recurrence/progression was associated with LOH (*p* = 0.0179). Our results suggest LOH as a genetic predictor of shorter progression-free survival, particularly within ependymoma and glioblastoma tumor types. Further genomic research with larger multi-institutional datasets should focus on these mutations as possible prognostic factors.

## 1. Introduction

Intramedullary spinal cord tumors (IMSCTs) comprise a rare but diverse group of tumors with associated variable outcomes. They comprise two to five percent of spinal tumors and lead to invasion of the gray and white matter [1]. The most common IMSCTs by decreasing incidence are ependymomas, astrocytomas, and hemangioblastomas [2,3]. Other types of tumors include lipomas, germ cell tumors, gangliogliomas, germinomas, lymphomas and metastasis [1]. Clinically, symptoms depend on the location of the tumor in spinal cord and include sensorimotor deficits, myelopathy, proprioceptive deficits, and localized neck or back pain. Magnetic resonance imaging (MRI) is the method of choice to identify these tumors [1]. For most IMSCTs, maximal safe surgical resection and, if appropriate based on pathology, adjuvant radiation and chemotherapy remain the mainstays of treatment [4,5].

The diversity of the IMSCTs complicates the selection of specific therapeutic interventions. Several case series of IMSCTs in single-center studies have been recently reported in the literature [4,5]. However, those studies focus on the surgical management of the tumor and do not include molecular characterization or genetic analysis. Furthermore, though diagnosis and management of supratentorial tumors is supported by molecular characterization, a similar approach to IMSCTs has not been effectively defined.

Recent reports have highlighted emerging unique molecular and genetic features of IMSCTs [6]. Early evidence suggests that IMSCTs harbor unique genetic profiles compared to intracranial tumors that may explain differences in prognosis and better guide therapy [6,7,8,9,10]. Unfortunately, characterization of these genetic profiles is limited. Here, we describe the largest reported genetic characterization of IMSCTs in a cohort of patients as well as the utility of copy number changes for prognostication and management.

## 2. Materials and Methods

A descriptive retrospective analysis of all patients undergoing resection of intramedullary spinal cord tumors at Emory University and associated hospitals from 1999 to 2020 was performed. Institutional Review Board approval was obtained for patient record analysis, and the need for consent was waived. A total of 93 IMSCT patient records were reviewed, and 61 records fit the inclusion criteria of availability of pre- and post-operative imaging, physical exams/short-term follow-up, and pathology report. Of the 61 records reviewed, a total of fourteen (14) patient’s pathology specimens had either a SNaPshot mutation panel or OncoScan SNP-CN array. Eight (8) patients had specifically undergone whole-genome sequencing via an OncoScan SNP-CN array. The decision to undertake sophisticated molecular investigations in patients with intramedullary spinal cord tumors stemmed from the neuropathologist’s imperative to augment diagnostic precision by acquiring Appendix A.

A SNaPshot mutation panel was performed using the multiplexed PCR-based assay (SNaPshot) to simultaneously identify 44 mutations in 10 genes (listed below). This test detects heterozygous mutations in tissue with more than 20% tumor content or homozygous mutations in tissue with more than 10% tumor content. The genes studied were AKT1, BRAF, EGFR, IDH1, IDH2, KRAS, MEK1, NRAS, PIK3CA, and PTEN5.

An SNP Copy Number (SNP-CN) array analysis was performed using the Thermo Fisher Scientific OncoScan FFPE Assay Kit (Thermo Fisher, Waltahm, MA, USA) on genomic DNA isolated from formalin-fixed paraffin-embedded (FFPE) tissue. The OncoScan platform queried 239,038 markers (19,038 non-polymorphic markers and 220,000 SNP markers, with increased density within approximately 900 cancer or cancer-related genes) and includes detection of copy number abnormalities, gene deletion and amplification events as well as loss of heterozygosity and allelic imbalances across the entire human genome [8,9,10,11,12,13,14,15,16]. Data analysis was performed using the Thermo Fisher Scientific CHAS 4.1 software (Waltham, MA, USA) and OncoScan Nexus Express software and aligned to the National Center for Biotechnology Information (NCBI, Bethesda, MD, USA) human build GRCh38 assembly. The OncoScan assay utilized molecular inversion probe technology which is optimized for FFPE samples.

## 3. Statistical Analysis

Descriptive statistics for each variable were calculated. Univariate analyses were performed to assess factors associated with progression-free survival (PFS) using the log-rank test and Cox proportional hazards model for p53 expression, BRAFV600E mutation, KRAS mutation, gene gain, gene loss, and copy neutral loss of heterozygosity. PFS was defined as the time from the date of surgery to the date of tumor recurrence or death, whichever occurred first. Firth’s penalized likelihood approach was used to account for the low event rates. Fisher’s exact test was performed to compare whole-genome analyses and specific gene mutations with progression. For this analysis, only binary data (for gene loss count, gene gain, copy neutral loss of heterozygosity, total number of genetic alterations, p53 overexpression, and ATRX deletion) were evaluated.

All statistical tests were two-sided, with a *p* value < 0.05 considered statistically significant. All the above statistical analyses were performed using SAS Version 9.4 (SAS Institute, Inc., Cary, NC, USA). Circos plots were generated via the Circo software, using the whole-genomic data available from the eight (8) patients with the OncoScan SNP array [17]. Lastly, a Kaplan–Meyer curve was generated using GraphPad Prism.

## 4. Results

### 4.1. Characteristics of Patients and Tumors

Most patients were female (57.4%), and the average age at surgery was 47 (±15) years old (Table 1). Most tumors were located in the cervical region (60.7%). The extent of resection was the gross total resection (GTR) in 25 tumors (41%) and >75% in 24 (39.4%). Mean length of the follow-up was 3.5 years (range = 0–19 years). Not surprisingly given the location of these tumors, new acute focal neurological deficit was seen in 19 patients (31%), but most of these deficits were not permanent, as evidenced by the return to the pre-operative baseline in 84% (52 patients) at the time of the last office visit. In this cohort of 61 patients, the majority (49.1%) of tumors were ependymomas (Table 2).

### 4.2. Genetic Analysis

Fourteen patients had genetic testing performed on their pathology specimens, including the SNaPshot mutation panel and OncoScan SNP-CN array (Appendix A). The neuropathologist opted to conduct either the SNaPshot mutation panel, the OncoScan SNP-CN array, or both based on a strategic assessment of the diagnostic requirements for each individual case. By leveraging such targeted molecular analyses, the neuropathologist aimed to extract comprehensive insights into the underlying genetic alterations, facilitating a more refined and accurate diagnosis. The choice to perform one study, the other, or a combination thereof was dictated by the nuanced nature of the tumor’s molecular profile and the imperative to tailor the diagnostic approach to each patient’s unique circumstances at the time of diagnosis. Six patients had only the SNaPshot mutation panel, four (4) patients had only the OncoScan SNP-CN array, and four (4) had both tests performed. Using Circa (OMGenomics v. 1.2.2) a Circos plot was created for the eight patients who had whole-genome sequencing via the OncoScan SNP-CN array. In this plot, each circle represents a distinct pathology, with the blue myxopapillary ependymoma being the innermost circle, and the dark gray glioblastomas being the outer circles. The ependymomas compose the circles in between. There were five (62.5%) patients with ependymomas, two (25%) patients with glioblastomas, and one (12.5%) patient with myxopapillary ependymoma (Figure 1). Within each circle lies a colored region of the chromosome that was found to be altered. For example, the red regions indicate a deletion, the green regions indicate a gain of function, and the yellow indicates a loss of heterozygosity. This plot permits a clear, comparative visualization of the different pathologies and genetic alterations in these patients.

DeletionGainLoss of HeterozygosityGlioblastomaEpendymomaMyxopapillary Ependymoma

The univariate analyses to assess the influence of genomic mutations on PFS using the Cox proportional hazards model did not show statistical significance for copy number gain, copy number loss, p53 overexpression, BRAFV600E mutation, or KRAS mutation. However, it did show statistical significance for copy neutral loss of heterozygosity (*p* = 0.008) (see Table 3). Additionally, analysis with Fisher’s exact test showed a statistically significant correlation with copy neutral loss of heterozygosity and tumor progression (*p* = 0.0179). Copy number loss, total number of mutations, p53 overexpression, and ATRX mutation trended towards significance; however, copy number gain did not (see Table 4).

The Kaplan–Meyer curve (Figure 2) shows PFS in patients with LOH as opposed to non-LOH, as measured in percent survival (%) over time (months) (Figure 2). The analysis shows that patient survival probability drops from 65% to 35% to 0% in the timeframe of 20 months. There were no survivors past 20 months in the loss of heterozygosity group. Meanwhile, the non-loss of heterozygosity group remains at 100%, with no stepwise decline in survival probability.

## 5. Discussion

To our knowledge, this is the first study to identify copy neutral loss of heterozygosity as a predictor of progression-free survival in patients with IMSCTs. Both somatic genetic and epigenetic processes contribute to the development of cancer. Loss of heterozygosity frequently contributes to tumorigenesis due to the loss of tumor suppressor genes, resulting in an inactivated allele left in its genome [18,19]. Similarly, *MYC*-amplified spinal cord ependymoma has consistently been shown to have inactivating mutations and loss of heterozygosity of the *NF2* gene, which correlates with unfavorable outcomes [20].

IMSCTs are rare and diverse and present unique clinical challenges in part because of their biological intricacies. Though tumor surgical resection remains a mainstay of treatment for IMSCTs, adequate interrogation of certain molecular underpinnings could improve patient stratification, prognostication, and management. DNA copy number changes and correlation to clinical outcome have been studied in intracranial tumors, but similar data does not exist for IMSCTs [21,22]. Using the 21-year registry of IMSCTs at our institution, we specifically examined DNA copy number changes and their impact on clinical outcomes and survival.

Pajtler and collaborators reported a clear genetic distinction between intracranial and spinal ependymomas [13]. Zhang et al. further delineates the genetic difference between IMSCTs and their brain counterparts [19]. While studying the biologically distinct IMSCTs, specific gene mutations have been suggested as useful outcome predictors, but their correlation with clinical outcomes is still debated. For example, while KIAA1549-BRAF and BRAFV600E mutations have shown to correlate with better outcomes in pediatric intramedullary low-grade gliomas [23,24], their impact on adult intramedullary gliomas is not clear [25]. Consistent with the latter report, our data in adults also failed to show any impact on PFS in patients with the BRAFV600E mutation.

Interestingly, our results suggest that copy neutral loss of heterozygosity is a significant genetic event that correlates with a shorter progression-free survival, particularly with respect to ependymomas and glioblastomas (Figure 2, Table 5). This has not been previously reported. Though not statistically significant, tumors harboring *ATRX* and *TP53* mutations did demonstrate a trend towards shorter PFS. Both mutations have been reported in spinal cord high-grade astrocytomas and glioblastomas and are associated with recurrence and shorter progression-free survival [26,27,28]. The expected mutations which affect PFS (p53 and ATRX) did not show a significant association.

The Circos plot (Figure 1) showed no clear preponderance towards a specific mutation in our sample of tumors. However, several mutations did appear repeatedly across multiple samples, and the chromosome with the most genetic variations was chromosome 22. This was expected because most of the patients analyzed had intramedullary ependymomas (5–62.5%) and mutations in chromosome 22 are well known in patients with and without NF-2 with spinal ependymomas (SE) [20]. We could not find previous reports analyzing the impact of LOH on chromosome 22 in patients with SE. Our results demonstrate that copy neutral LOH events in the presence of spinal cord glioblastoma and ependymoma correlate with poorer outcomes, specifically PFS. Our data expand on the known body of work related to mutations in chromosome 22 in SE and suggest a shorter PFS when there is copy neutral LOH on this chromosome [21,22]. In our dataset, only patients with copy neutral LOH and mutations on chromosome 22 experienced rapid local recurrence and death. This once again suggests an effect of copy neutral LOH as a significant predictor of survival.

Future studies should consider analyzing chromosomal mutations and the development of targeted agents to this specific DNA copy number change. As an example, our two (2) cases of intramedullary glioblastoma (iGBM) can be compared using the Circos plot (Figure 1). The plot shows us that they are genetically distinct from their intracranial glioblastoma (cGBM) and exhibit LOH in two chromosomes not commonly seen in cGBM [21,25,27,28]. Previous publications have shown LOH in cGBM in chromosomes 10 and 19 [29,30]. However, these changes in DNA copy numbers were not seen in our iGBM cohort that exhibited LOH in chromosomes 9 and 22. LOH in chromosome 9 has been reported in spinal pilocytic astrocytomas, and it is typically associated with CDKN2A tumor suppressor gene mutations but has not been associated with prognosis [7]. Conversely, there are a few publications associated with LOH in chromosome 22 and prognosis. Specifically, LOH has been seen in association with the NF-2 gene (merlin) but has not been correlated with prognosis in sporadic intramedullary tumors which are not associated with NF-2. Additionally, LOH in chromosome 22 has been associated with poor prognosis in sporadic extramedullary meningiomas and schwannomas with a higher recurrence rate and proliferation index [31,32]. The finding of LOH in chromosome 22 (22q11.1q13) in two out of the three patients who progressed in our cohort is an indication that more research should be conducted on the clinical implications of these mutations in IMSCTs.

### Limitations

It is important to acknowledge the limitations of the present study. Although the cohort of IMSCTs is large and compares with the sample size in other series [5], only a subset of patients underwent genetic analysis and genome sequencing. This is due to the rarity of these neoplasms and the fact that the technology to perform this analysis has only recently been developed. We focused on specific gene mutations and DNA copy number changes but did not look at epigenetic or expression-based changes. More recently, the application of methylation array technology to further characterize CNS neoplasms has proven to be a robust and powerful tool to define the subtypes of CNS neoplasms [33,34,35,36].

## 6. Conclusions

In the present study, we performed a retrospective analysis of IMSCTs and DNA copy number changes and their impact on clinical outcomes. Our results demonstrate that copy neutral loss of heterozygosity (LOH) is a statistically significant genetic event which suggests a shorter progression-free survival, particularly in spinal cord ependymoma and glioblastoma. Of interest, LOH in chromosome 22 (22q11.1q13) was seen in most patients with whole-genome sequencing who progressed after treatment. Due to the limited number of cases, future studies should be conducted looking at LOH mutations in patients diagnosed with IMSCTs. As genomic technologies become more accessible and cost-effective, the integration of comprehensive genetic profiling into clinical practice has the potential to revolutionize our understanding and treatment of these tumors. The focus on LOH mutations, as elucidated by our study, should serve as a rallying point for directing future genetic studies that could enhance diagnosis, treatment and prognosis of patients with IMSCTs.

## Figures and Tables

**Figure 1 cancers-16-00404-f001:**
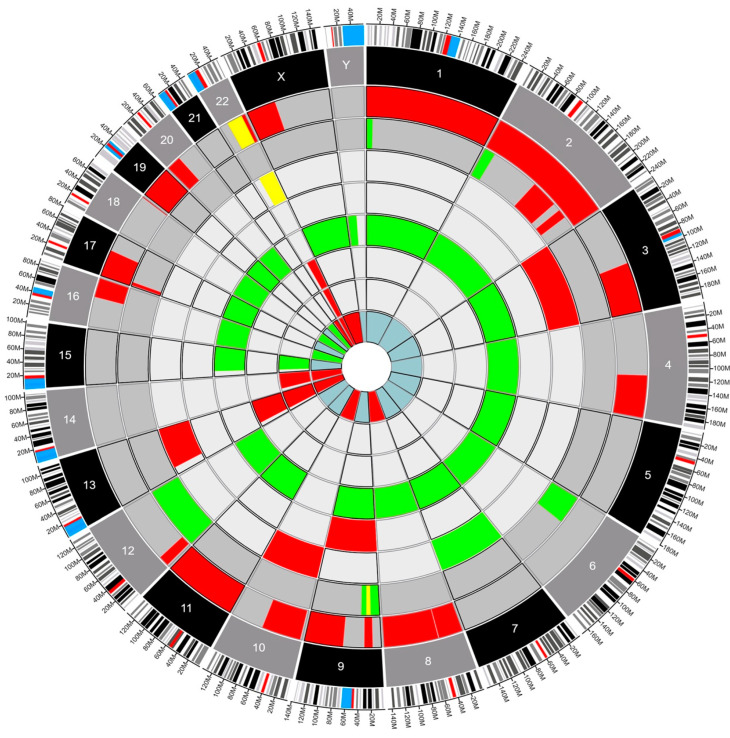
Circos plot depicting structural variants in eight IMSCTs selected for whole-genome sequencing. Red, deletion. Green, gain. Yellow, loss of heterozygosity. Dark gray, glioblastoma. Gray, ependymoma. Blue, myxopapillary ependymoma.

**Figure 2 cancers-16-00404-f002:**
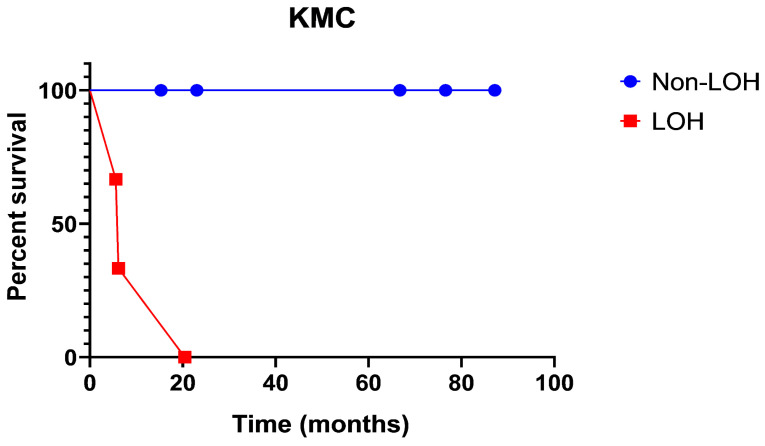
Progression-free survival in patients with loss of heterozygosity as opposed to non-loss of heterozygosity, as measured in percent survival (%) over time (months).

**Table 1 cancers-16-00404-t001:** Demographics.

Variable	Level	*n* (%) = 61
Sex	Female	35 (57.4)
	Male	26 (42.6)
Age at surgery	Mean	47
	Median	46
	Minimum	21
	Maximum	76
	Std Dev	15
Hospital days	Mean	8
	Std Dev	4.47
Tumor location	Cervical	37 (60.7)
	Cervicothoracic	1 (1.6)
	Lumbar	5 (8.2)
	Thoracic	17 (27.9)
	Thoracolumbar	1 (1.6)
Resection %	Biopsy	6 (9.8)
	100	25 (41.0)
	50–78	4 (6.6)
	78–99	20 (32.8)
	<50	6 (9.8)
Outcomes	Mortality	1 (1.6)
	Morbidity	19 (31.2)
KPS	Mean	77
	Std Dev	14
Post Op KPS	Mean	71
	Std Dev	16
FU Total Days	Mean	1287
	Median	721
	Minimum	4
	Maximum	6819
	Std Dev	1600

FU = Follow-up; KPS = Karnofsky performance scale.

**Table 2 cancers-16-00404-t002:** Pathology.

Variable	Level	*n* (%) = 61
Pathology	Ependymoma	24 (39.3)
	Hemangioblastoma	14 (23.0)
	Myxopapillary Ependymoma	5 (8.2)
	Low grade astrocytoma	4 (6.6)
	Anaplastic Astrocytoma	2 (3.3)
	Glioblastoma	2 (3.3)
	Lipoma	2 (3.3)
	Mature Teratoma	2 (3.3)
	Anaplastic Ependymoma	1 (1.6)
	B cell lymphoma	1 (1.6)
	Benign Neuroepithelial Cyst	1 (1.6)
	Cavernous Hemangioma	1 (1.6)
	Metastatic Carcinoma	1 (1.6)
	Schwannoma	1 (1.6)

**Table 3 cancers-16-00404-t003:** The univariate analyses to assess factors associated with progression-free survival using the Cox proportional hazards model.

			PFS (Mths)	
Covariate	Level	N	Hazard Ratio (95% CI)	*p* Value
Gain	No	3	0.44 (0.01–5.49)	0.343
	Yes	5	-	-
Loss	No	1	1.09 (0.01–13.45)	0.545
	Yes	7	-	-
Copy Neutral Loss of Heterozygosity	No	5	0.05 (0.01–0.69)	0.008
	Yes	3	-	-
p53 status.	Not Over-expressed	5	0.29 (0.02–3.58)	0.351
	Over-expressed	2	-	-
BRAFV600E	Negative	9	0.92 (0.07–126.63)	0.545
	Positive	1	-	-
KRAS results	Negative	6	0.11 (0.00–2.08)	0.083
	Positive	1	-	-

Firth’s penalized maximum likelihood estimation was used.

**Table 4 cancers-16-00404-t004:** Fisher’s exact test comparing SNaPshot mutation panel results and OncoScan SNP-CN array results with tumor progression.

		Group		
Covariate	Level	Not Progressed	Progressed	*p* Value *
Loss Count	≤3	4	0	0.143
	>3	1	3	
Copy Neutral	No	5	0	0.0179
	Yes	0	3	
Total	≤8	3	0	0.196
	>8	2	3	
p53	Not Over-expressed	4	1	0.143
	Over-expressed	0	2	
ATRX	Not Deleted	3	0	0.1
	Deleted	0	2	
Gain Mutation	No	2	1	1
	Yes	3	2	

* The *p* value was calculated using Fisher’s exact test.

**Table 5 cancers-16-00404-t005:** Comparison of the survival curves.

Comparison of Survival Curves		
Log-rank (Mantel–Cox) test		
Chi square	7.647	
df	1	
*p* value	0.0057	
*p* value summary	**	
Are the survival curves sig different?	Yes	
Gehan–Breslow–Wilcoxon test		
Chi square	6.759	
df	1	
*p* value	0.0093	
*p* value summary	**	
Are the survival curves sig different?	Yes	
Median survival		
Dataset-A	Undefined	
Dataset-B	6.143	
Hazard Ratio (Mantel-Haenszel)	A/B	B/A
Ratio (and its reciprocal)	0.02802	35.68
95% Cl of ratio	0.002224 to 0.3531	2.832 to 449.6
Hazard Ratio (logrank)	A/B	B/A
Ratio (and its reciprocal)	0.000	
95% Cl of ratio	−1.000 to −1.000	−1.000 to −1.000

## Data Availability

Data could be shared with interested parties by contacting interested investigators.

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
