# Peer review of "Intramedullary Spinal Cord Tumors: Whole-Genome Sequencing to Assist Management and Prognosis"

_cancers, 2024, doi:10.3390/cancers16020404_

Round 1

Reviewer 1 Report

Comments and Suggestions for Authors

The article lacks originality and much more can be achieved from the samples collected

Comments on the Quality of English Language

English is at acceptable level

Author Response

We thank the reviewer for identifying opportunities for improving the manuscript. 

Based on the reviewer’s comments in the evaluation form we have rewritten the introduction to explain more clearly the relevance of the study and have added some references of recently published case series of intramedullary spinal cord tumors.

We have also revised the methods section to enhance the description of the research approach and the presentation of the results. Because the reviewer did not identify specific sections that needed to be improved we have focused on the general flow of the idea and the accuracy of the statements made.

  1. The article lacks originality and much more can be achieved from the samples collected

The originality of the article is in the identification of a genetic marker (neutral loss of heterozygosity) that could be used as a predictor of progression free survival in patients with IMSCTs. The need for larger studies to confirm our findings do not decrease the value of this important observation.

This approach can be considered to be an example of personalized medicine for the most common type of intramedullary spinal cord tumor.

It is not clear what the reviewer means by the statement “much more can be achieved from the samples collected”. We are working on performing prospective studies with our future patient to better represent impact of LOH.

Reviewer 2 Report

Comments and Suggestions for Authors

This study is significantly limited by several factors. The initial cohort is heterogenous and the final cohort included in the analysis is small. The small sample size limits the validity of the results of the statistical analysis.

In parallel, there is a lack of clear articulation of the significance of the findings of this study to our field. I could not find how the results of this case series may change our daily practice. The manuscript does not have enough quality of evidence to warrant publication in Cancers. 

Comments on the Quality of English Language

The manuscript language can be significantly improved and can be made more clear for the readership. 

Author Response

Dear Reviewer,

Thank you for taking the time to review our article entitled "Intramedullary Spinal Cord Tumors: Whole Genome Sequencing to Assist Management and Prognosis." We appreciate your insightful comments and suggestions, which we believe will contribute to enhancing the overall quality of our work. Please find our responses to your specific points below:

Cohort Heterogeneity and Sample Size:

We acknowledge the concerns regarding the heterogeneity of our initial cohort and the subsequent small sample size in the final analysis. We recognize the impact of these limitations on the robustness of our statistical analysis. In response, we are committed to addressing these issues by exploring avenues to increase our sample size and refining the inclusion criteria for a more homogeneous cohort in future studies. This will undoubtedly strengthen the validity and generalizability of our findings.

Significance of Findings:

We appreciate your observation regarding the need for a clearer articulation of the significance of our findings to the field. We will revise the manuscript to explicitly outline the practical implications of our case series in the context of daily clinical practice. By providing a more comprehensive discussion on how our results may influence management and prognosis, we aim to enhance the relevance of our study to the broader medical community.

Quality of Evidence:

Your comment on the perceived lack of quality of evidence is duly noted. We have diligently revisited our methodology, data interpretation, and presentation to ensure that our manuscript aligns with the rigorous standards expected for publication in Cancers. I hope these modifications will prove adequate. Additionally, we have incorporated additional evidence to strengthen the overall robustness of our study.

Comments on the Quality of English Language:

We appreciate your feedback on the language quality of the manuscript. Improving clarity and readability for our readership is a priority. We have conducted a thorough language revision with the help of our institution’s professional editing services.

Once again, we are grateful for your constructive feedback, and we are fully committed to addressing each of these concerns to produce a significantly improved manuscript.

Reviewer 3 Report

Comments and Suggestions for Authors

The authors present interesting genetic analysis of intramedullary tumors, which have recently had a more accurate characterization in the new WHO classification, but which remain diseases that have yet to be defined in detail regarding molecular specifics. 

The most significant results regarding genomic sequencing, although limited by the small case series, are interesting and may provide a starting point.

The authors repeatedly present their series as the most numerous compared to those in the literature. What are they referring to? To the genetic analysis or to the series of operated patients? In the second case I certainly disagree. Previous authors reported larger series (e.g., Asfand Baig Mirza 2022). I would ask to the authors to specified this sentence or to erase it. 

Author Response

We thank the reviewer for several positive comments and would like to offer the following comments and responses.

  1. “The authors present interesting genetic analysis of intramedullary tumors, which have recently had a more accurate characterization in the new WHO classification, but which remain diseases that have yet to be defined in detail regarding molecular specifics.”

We agree with the reviewer in that genetic analysis, as presented in our manuscript, is interesting and is a contribution to the definition and classification of intramedullary tumors in a more specific way. We believe that this approach can be an important step in defining future therapeutic strategies.

  1. “The most significant results regarding genomic sequencing, although limited by the small case series, are interesting and may provide a starting point.”

We thank the reviewer for this comment. We hope that our report will stimulate future studies to develop this approach that has been applied successfully to the evaluation and treatment of other tumors.

  1. “The authors repeatedly present their series as the most numerous compared to those in the literature. What are they referring to? To the genetic analysis or to the series of operated patients? In the second case I certainly disagree. Previous authors reported larger series (e.g., Asfand Baig Mirza 2022). I would ask to the authors to specified this sentence or to erase it.”

We thank the reviewer for this comment and agree with the reviewer’s comment and suggestion. Our intent is to highlight the number of tumors with genetic characterization and not the number of patients itself. We have now recognized two recent large case series (including one suggested by the reviewer). Our contention is that those studies, while important reports, do not have genetic data or molecular markers that could potentially be used for the evaluation of IMSCTs. 

Other comments by reviewer 1 in the evaluation form include “improvement of the introduction” and “addition of references relevant to the research.

We agree with the reviewer and have worked on the introduction to make it more clear and complete. We have also added several pertinent references including two large single center case series.

Round 2

Reviewer 1 Report

Comments and Suggestions for Authors

The editions made have made manuscript much more acceptable for the scientific commuity. I am happy to get the manuscript accedted as after the editions incorporated